# Non-Precious Metal Alloy Double Crown-Retained Removable Partial Dentures: A Cross-Sectional In Vivo Investigation

**DOI:** 10.3390/ma15176137

**Published:** 2022-09-04

**Authors:** Malin Strasding, Samir Abou-Ayash, Thomas Laziok, Sam Doerken, Ralf-Joachim Kohal, Sebastian Berthold Maximilian Patzelt

**Affiliations:** 1Division of Fixed Prosthodontics and Biomaterials, University Clinic of Dental Medicine, University of Geneva, Rue Michel-Servet 1, 1211 Geneva, Switzerland; 2Department of Reconstructive Dentistry and Gerodontology, University of Bern, Freiburgstrasse 7, 3007 Bern, Switzerland; 3Privat Practice, 79104 Freiburg im Breisgau, Germany; 4Institute of Medical Biometry and Statistics, Medical Center, University of Freiburg, 79104 Freiburg im Breisgau, Germany; 5Center for Dental Medicine, Department of Prosthetic Dentistry, Faculty of Medicine, University of Freiburg, Hugstetter Str. 55, 79106 Freiburg im Breisgau, Germany; 6Private Dental Clinic, Am Dorfplatz 3, 78658 Zimmern ob Rottweil, Germany

**Keywords:** removable partial denture, removable dental prosthesis, non-precious metal alloy, telescopic crown, patient satisfaction, patient-reported outcomes

## Abstract

(1) Background: An alternative material to precious metal alloys are non-precious metal alloys. The material properties of these are different and, therefore, their clinical, biological and mechanical behaviors may also differ. Hence, the purpose of this in vivo investigation was to analyze the clinical and patient-reported outcomes of patients restored with non-precious metal alloy double crown-retained removable partial dentures (NP-D-RPDs). (2) Methods: Partially edentulous patients were restored with non-precious metal alloy partially veneered NP-D-RPDs. Survival rates, success rates, failures and patient-reported outcomes were investigated and statistically evaluated. (3) Results: A total of 61 patients (65.6 ± 10.8 years) were included and clinically and radiographically examined. The mean follow-up time was 25.2 ± 16.5 months. In total, 82 NP-D-RPDs and 268 abutment teeth were examined. The overall survival rate of the NP-D-RPDs was 100% after a mean follow-up time of 2.1 years. The overall success rate was 68.3%. The overall satisfaction with the NP-D-RPDs was 94.3%. (4) Conclusions: Non-precious metal alloy partially veneered NP-D-RPDs seem to be an efficient alternative to precious metal alloy RPDs with excellent patient-reported outcomes.

## 1. Introduction

Worldwide, people are confronted with tooth loss at a certain time in life. Even though considerable success in maintaining teeth has been achieved with oral prevention strategies, caries and periodontitis are still prevalent and can lead to the loss of teeth [1]. Over recent years, a shift from treating completely edentulous patients to treating partially edentulous patients has occurred [2,3]. Regarding the treatment of patients suffering from severe tooth loss, clasp-retained removable partial dentures (RPDs) with metal frameworks are the standard treatment in most countries. Nevertheless, these types of RPDs are accompanied by an increased risk of caries, esthetic impairment, and unfavorable load on the denture-bearing tissues and abutment teeth [4,5,6]. 

Double crown-retained RPDs represent an alternative treatment to clasp-retained RPDs, and are frequently applied, especially in Germany and Japan. Generally, double crowns can be divided into two types: telescopic crowns, with parallel or nearly parallel walled (0–2°) primary crowns, and conical crowns, with tapered primary crowns (taper ≥ 3°). The retention force of double crown-retained removable partial dentures (D-RPDs) mainly depends on the number and length of parallel surfaces, whereas the retention of conical crown-retained RPDs is determined by the taper and the joining force [7,8,9]. The advantages of these compared to clasp-retained dentures are improved esthetics, a better long-term prognosis of abutment teeth and dentures, a decreased risk for caries, easy extensibility in case of abutment tooth loss, and axial force distribution on the abutment teeth [9,10,11].

The most frequently employed materials for the fabrication of primary and secondary crowns in D-RPDs are precious metal alloys, leading to sufficient long-term retention [11,12]. However, high material costs are the major disadvantage of precious metal alloys. Several approaches have been made to replace precious metal alloys with other materials, including non-precious alloys, high strength ceramics (ZrO_2_), and high-performance polymers [13,14]. Both in vivo and in vitro studies focusing on these materials have revealed promising results [14,15,16,17]. Particularly, non-precious alloys have been used for the fabrication of primary and secondary crowns for several years; yet, little is known about the related clinical outcomes. Recently published retrospective studies on the clinical outcomes of non-precious alloy telescopic crown-retained RPDs demonstrated clinically acceptable results [18,19,20]. However, none of the studies reported on patient-reported outcomes (PROs). 

PROs are more and more frequently used in dentistry to evaluate the patients’ subjective ratings of a certain treatment outcome [20]. Various tools exist for the assessment of PROs such as patient interviews or questionnaires. One of the most frequently applied questionnaires is the visual analogue scale (VAS), as it represents a generic, multidimensional, and easily understandable instrument. However, PROs for RPD wearers have rarely been evaluated [21,22]. To the best of the authors’ knowledge, there is only one study available reporting on PROMs in D-RPDs with precious metal alloy primary crowns [23]. 

The present cross-sectional clinical study aimed to analyze the clinical and patient-reported outcome measures restored with non-precious metal alloy telescopic crown-retained removable partial dentures (NP-D-RPDs). Furthermore, factors affecting the outcomes were evaluated.

## 2. Materials and Methods

### 2.1. Study Design and Population

For the present cross-sectional study, all patients that had received NP-D-RPDs between 2009 and 2016 at the Department of Prosthetic Dentistry, Center of Dental Medicine, University of Freiburg, Germany, were invited to participate in the present retrospective examination. These patients had been treated by undergraduate students who were constantly supervised by the same four specialized prosthodontists. The supervisors had undergone calibration prior to the student education course and adhered to a clearly defined protocol established by the head of the department. All key intermediate steps during the treatment had to be validated by the supervisors. The study protocol was conducted in accordance with the guidelines of the Declaration of Helsinki and was approved by the local ethics committee (Albert-Ludwigs-University Freiburg, Germany; Application number: 338/15; date of approval: 24 November 2015). The study was registered at the German Clinical Trials Register (DRKS ID: DRKS00010075). All patients were informed in detail about the study’s purpose and procedures. Informed consent was obtained from all participating patients. 

### 2.2. Inclusion Criteria

Patients were provided with non-precious metal alloy NP-D-RPDs either for the maxilla, mandible or both jaws by undergraduate students as part of a pre-graduate prosthodontic training course. A minimum of two abutment teeth as retention for the denture and an existing clinical and radiographic final report at the time of insertion of the denture were essential. Further, patients had to be above 18 years of age.

### 2.3. Exclusion Criteria

Patients with NP-D-RPDs and/or primary crowns made out of precious metal alloys or zirconia were excluded. Additionally, patients with fewer than two abutment teeth or with partially or fully implant-retained NP-D-RPDs were not included in the study.

### 2.4. Tooth Preparation and Fabrication of the NP-D-RPDs

Abutment teeth were prepared with diamond burs with a pronounced rounded chamfer of 1.0 mm to 1.2 mm (Komet, Gebr. Brasseler GmbH & Co. KG, Lemgo, Germany) to achieve sufficient space for the primary and secondary crowns. All devital abutment teeth had undergone sufficient endodontic treatment prior to the prosthetic restoration. Endodontically treated abutment teeth always received a prefabricated or a cast metal post and core build-up (ER-Stabilisierungsstift or ELD-Stift, Komet, Gebr. Brasseler GmbH & Co. KG, Lemgo, Germany), depending on the amount of loss with regard to tooth structure [24]. The metal posts were cemented with an adhesive resin cement (Panavia 21, Kuraray Noritake, Kuraray Europe GmbH, Hattersheim, Germany), and direct core reconstructions were performed with a self-curing composite core build-up material (Clearfil core, Kuraray Noritake, Kuraray Europe GmbH, Hattersheim, Germany). 

The laboratory technician cast the non-precious metal alloy primary crowns. Afterwards, the primary crowns obtained a 0–2° parallel milling, resulting in an axial wall thickness of 0.2–0.3 mm. The secondary structure and framework were also manufactured out of a non-precious metal alloy, and connected by means of the one-piece casting method, laser welding or soldering. Secondary crowns were fully or partially veneered with a veneering resin composite (Material, Firma, Stadt, Land). The primary crowns were cemented either with a glass-ionomer cement (Ketac Cem, 3M ESPE Deutschland GmbH, Neuss, Germany) or, in the case of very short and conically prepared abutment teeth, with an adhesive resin cement (Panavia 21, Kuraray Europe GmbH, Hattersheim, Germany). After the insertion of the dentures, the fit and occlusion were tested, and, if necessary, adjusted. A sequential guidance of the canines and premolars for the RPDs was established. 

The RPDs were manufactured by 7 different laboratories, using 6 different non-precious metal alloys, all based on cobalt and chromium as main components without additional retentive structures. 

One week after denture delivery (baseline), all restorations were clinically evaluated. Within this evaluation, the marginal adaptation was verified with a clinical dental probe, and cement excess was controlled and eliminated when present. The pocket probing depth PPD (Michigan-o-Probe, Hu-Friedy, Frankfurt, Germany), Sulcus Bleeding Index (SBI) [25,26], and Plaque Index (PI) [27] of all remaining teeth were checked. Additionally, abutment tooth sensitivity and percussion were examined, and peri-apical radiographs of the abutment teeth were obtained. Patients were enrolled in a 6-month interval.

### 2.5. Clinical Examinations at the Present Follow-Up

All patients were clinically and radiographically examined. During the clinical examination, the following parameters were documented: the extent of the composite resin veneering (fully or facially veneered), number of abutment teeth, abutment tooth sensitivity to cold pulp sensitivity test, abutment tooth mobility (0–3 degrees) [26]. Sulcus Bleeding Index SBI [25], Plaque Index (PI) [27], and pocket probing depth (PPD) were registered. The marginal adaptation was checked with an intraoral dental mirror and a dental probe (Henry Schein Dental Deutschland GmbH, Langen, Germany). Shimstock metal foil (8 µm) (Hanel Shimstock Foil, Coltene, Altstätten, Switzerland) was used to check the occlusal contacts. The stability of the dentures was rated as good/medium/insufficient after the bi-manual loading and verification of the fit/retention in vertical, horizontal and sagittal directions. If the dentures did not move horizontally or vertically, the stability was as rated good. In case of slight horizontal or vertical rocking during bi-manual loading, the stability was rated as medium. Slight horizontal or vertical rocking was defined as a loss of retention, while the prosthesis did not loosen completely or drop down. If the dentures were levering out upon loading, the stability was rated as insufficient. Technical complications (loss of retention of the dentures, decementation of the primary crowns, framework fractures, fractures of the dentures or prosthetic teeth, and chipping of the veneering resin composite) were registered. Biological complications (secondary caries, loss of abutment tooth sensitivity, (apical) periodontitis, fractures of abutment teeth, loss of abutment teeth, adaptation/relining of the denture base due to sore spots and resorption of the alveolar ridge) were documented. All large defects (caries, fractures of the abutment teeth) were considered abutment tooth defects if they were clinically repairable, and the abutment tooth could therefore be maintained. The treatments comprised endodontic treatments, the insertion of a post and core build-up after abutment tooth fracture or periapical lesions, and composite fillings after secondary caries excavation. In addition to the clinical examination, information concerning possible additional maintenance visits and interventions was gathered by patient interviews and screening of the digital patient records since the delivery of the denture. Finally, radiographs of the abutment teeth were taken.

### 2.6. Patient-Reported Outcome Measures (PROMs)

Patient satisfaction was assessed by eight separate visual analogue scales (VAS), assessing the presence of pain, comfort, aesthetic appearance, function/chewing ability, retention/stability, phonation, cleanability and overall satisfaction with the NP-D-RPDs. It was further explained to the patients that “comfort” referred to wearing comfort, asking them to evaluate how comfortable the denture felt during function and in resting position (e.g., whether the patients felt any pressure spots that were not painful, yet uncomfortable), whereas “overall satisfaction” referred to the personal satisfaction of the patients when considering factors, such as comfort, aesthetics, function, cleanability, and other aspects. The patients were asked to mark how satisfied they were with their NP-D-RPDs 10 cm-long scales with the starting and endpoints representing scores of 0 and 100, respectively. Except for the presence of pain, where a score of 100 represented the worst possible outcome, all other scales were arranged with a score of 0 representing the worst outcome and 100 representing the best possible outcome.

### 2.7. Definitions

Follow-up time: Patients were divided into three groups depending on the follow-up time after insertion of the NP-D-RPDs. Short follow-up (SF) was defined as a 0- to 12-month follow-up. Medium follow-up (MF) was defined as a follow-up time of >12 to 36 months and patients that had received the NP-D-RPDs > 36 months ago were included in the long follow-up (LF) group. 

Success: NP-D-RPDs were provided without any technical complication until the follow-up examination. Additionally, no abutment tooth loss and no major, repairable defects occurred, such as abutment tooth fracture, which was repairable with post and core build-up. No adaptations of the denture were necessary and the denture stability was not rated as insufficient. 

Survival: NP-D-RPDs in situ and functioning at the time of the follow-up examination, even if the dentures had undergone repair or adaptation or tooth loss had occurred. Slight adjustments of the denture were accepted, as long as dentures had not been completely modified (e.g., extension of one tooth due to localized tooth loss). 

Failure: NP-D-RPDs were considered to have failed if all abutment teeth were lost since denture delivery, the denture could not be worn by patients due to misfit, or if denture retention was clinically unacceptable, for example, due to abutment tooth loss. NP-D-RPDs that could not be inserted at the time of re-examination or had completely lost their function were considered as failures. This included dentures which had been completely modified in the dental laboratory and were dissimilar to the initially inserted NP-D-RPD. (e.g., adjustment to a complete denture). Relining of the dentures due to alveolar ridge resorption and removal of sore spots during the first 6 weeks after insertion were not considered as failures. 

### 2.8. Statistical Analysis

For descriptive analyses, statistical frequency, mean and standard deviations were calculated. Boxplots were used for graphical representations of the results. Statistical significance of the categorical variables was verified by means of Pearson’s Chi Squared Test. For continuous variables (patient satisfaction), a one-way analysis of variance (ANOVA) was used.

The level of statistical significance was set to *p* < 0.05. The statistical analyses were performed with the software Stata (14.2, StataCorp LP, College Station, TX, USA).

## 3. Results

In the present cross-sectional study, 61 patients (age: 65.6 ± 10.8 years; age range: 39 y to 90 y) were clinically and radiographically examined in 2016. The mean follow-up time was 25.2 ± 16.5 months (minimum: 3 months; maximum: 78.9 months). In total, 82 NP-D-RPDs (62.2% male patients, 37.8% female patients) and 268 abutment teeth were examined. Seventeen patients with 21 NP-D-RPDs (25.6%) were included in the SF group. This comprised 8 maxillary dentures and 13 mandibular dentures. In total, 33 patients with 46 dentures (56.1%) were allocated to the MF group, out of which 23 dentures were inserted in both jaws. Fifteen patients with 15 dentures (18.3%) were included in the LF group. Out of these, nine dentures were located in the maxilla, and six dentures in the mandible. Four patients received their dentures at different time points, and were therefore included in two follow-up groups.

### 3.1. Survival and Success Rates of the NP-D-RPDs

The overall survival rate of the NP-D-RPDs was 100%. No NP-D-RPD was lost over the mean follow-up time of 2.1 years. The overall success rate was 68.3%. 

### 3.2. Outcome on Denture Level

Technical complications occurred in 22% (*n* = 18) of the dentures. Chippings of the veneered telescopic crowns were found in 9.8% (*n* = 8) of the dentures. In 4.9% (*n* = 4) of the dentures, a prosthetic resin tooth had to be renewed due to fracture or clinically unacceptable wear. Chipping and an additional prosthetic tooth fracture/wear in the same denture occurred in 4.9% (*n* = 4). Denture stability was rated sufficient in 87.8% (*n* = 72), moderate in 9.8% (*n* = 8) and insufficient in 2.4% (*n* = 2) of the cases.

Besides the technical complications, sore spots had to be eliminated in 3.7% (*n* = 3) of the dentures and 1.2% (*n* = 1) of the dentures needed relining. In 19.5% (*n* = 16) of the dentures at least one abutment tooth lost sensitivity. 

### 3.3. Outcome on Abutment Tooth Level

Out of a total of 268 abutment teeth, secondary caries was found in 4.1% (*n* = 11). In 5.6% (*n* = 15) of the abutment teeth, large defects (caries, fractures of the abutment teeth) had to be treated in order to maintain the abutment tooth. These treatments comprised endodontic treatment, followed by the insertion of a post and core build-up due to apical periodontitis or abutment tooth fracture, and/or composite fillings after secondary caries excavation. Overall, 66.8% (*n* = 179) of the abutment teeth reacted sensitive to the cold pulp sensitivity test, 16.8% (*n* = 45) of the abutment teeth were endodontically pre-treated, 3% (*n* = 8) showed a slight hypersensitivity, and 11.2% (*n* = 30) lost sensitivity throughout the follow-up period. During the follow-up time, 2.2% (*n* = 6) of abutment teeth were lost in four dentures. PPDs of > 4 mm were found in 20.5% (*n* = 55), a positive SBI was observed in 52.2% (*n* = 140) of the abutment teeth.

### 3.4. Results in the SF Group

For the SF group (21 dentures and 65 abutment teeth), with a mean follow-up period of 7.1 ± 3.4 months, the survival rate of the dentures was 100%, and the success rate was 95.2% (Table 1). No technical complications occurred in this group. No secondary caries was detected at the follow-up examination. The stability of all dentures was rated as clinically sufficient. Throughout the follow-up period, repairable abutment tooth defects (carious lesions, abutment tooth fractures) occurred in 3.1% (*n* = 2) of abutment teeth and in 4.8% (*n* = 1) of the dentures throughout the follow-up period. In total, 78.5% (*n* = 51) of the abutment teeth were found to be sensitive in the sensitivity test, 12.3% (*n* = 8) were endodontically pre-treated, 6.2% (*n* = 4) had lost sensitivity, and 3% (*n* = 2) showed slight hypersensitivity. 

### 3.5. Results in the MF Group

In the MF group (46 dentures and 152 abutment teeth), with a mean follow-up period of 24.9 months ± 6.9 months, the survival rate of the dentures was 100%, and the success rate was 63% (Table 1). Technical complications occurred in 26.1% (*n* = 12) of the dentures, 10.9% (*n* = 5) of which was due to chipping of the veneering composite resin, 6.5% (*n* = 3) renewal of resin denture teeth, and a combination of both chipping and renewal of resin denture teeth in 6.5% (*n* = 3). The stability of 2.2% (*n* = 1) of the denture was rated clinically insufficient.

Besides the technical complications, repairable abutment tooth defects (carious lesions and abutment tooth fractures) occurred in 8.7% (*n* = 4) of the dentures. Secondary caries was diagnosed in 5.3% (*n* = 8) of the abutment teeth. At follow-up examination, 65.1% (*n* = 99) of the abutment teeth were found to be sensitive in the cold pulp sensitivity test, 15.1% (*n* = 23) were endodontically pre-treated, loss of sensitivity was found in 13.8% (*n* = 21) and 4% (*n* = 6) showed slight hypersensitivity. In total, 2% (*n* = 3) of the abutment teeth had unrepairable lesions and therefore were lost in 4.3% (*n* = 2) of the dentures. 

### 3.6. Results in the LF Group

In the LF group (15 dentures and 51 abutment teeth), with a mean follow-up period of 51.7 ± 12.7 months, the survival rate of the dentures was 100% and the success rate was 46.7%. In total, 40.1% of the dentures (*n* = 5) exhibited technical complications (Table 2), 20% (*n* = 3) of which was chipping of the veneering composite resin, 6.7% (*n* = 1) renewal of resin denture teeth, and a combination of both, chipping and renewal of resin denture teeth in 6.7% (*n* = 1). Insufficient denture stability was found in 6.7% (*n* = 1). 

At the abutment tooth level, clinically repairable defects (carious lesions and abutment tooth fractures) were observed in 17.6% (*n* = 9) of the abutment teeth during the follow-up period. Secondary caries was detected in 5.9% (*n* = 3) of the abutment teeth at follow-up examination.

At the follow-up examination, 56.9% (*n* = 29) of the abutment teeth were found to be sensitive in the cold pulp sensitivity test, 27.4% (*n* = 14) were endodontically pre-treated, and loss of sensitivity was found in 9.8% (*n* = 5). In total, 5.9% (*n* = 3) of the abutment teeth were lost in 20% (*n* = 3) of the dentures due to unrepairable lesions during the follow-up period. 

### 3.7. Comparisons and Correlations

Kennedy Class I patients presented significantly higher success rates (32.1%; *n* = 18) than Kennedy Class II patients (7.1%; *n* = 4) (*p* = 0.003). Kennedy Class II patients presented significantly more dentures with abutment tooth loss (33.3%), compared to abutment tooth loss (7.9%) in the dentures of Kennedy Class I patients (*p* = 0.002). 

Dentures with sensitive abutment teeth had a success rate of 76.2%, dentures including endodontically pre-treated abutment teeth with post and core build-up had a success rate of 70.8%, and dentures retained by at least one non-sensitive abutment tooth without endodontic treatment showed a significantly lower success rate of 43.8% (*p* < 0.001).

Patients that exhibited secondary caries on abutment teeth had lower denture success rates (22.2%) compared to those with abutment teeth without secondary caries (74%; *p* = 0.002).

Patients with a high PI showed significantly more elevated PPD (>4 mm) and a positive SBI (*p* < 0.001) (Table 2). Additionally, significantly higher abutment tooth loss was registered in patients with high PIs (16.7%) compared to patients with low PIs (0%) (*p* = 0.035).

The sub-analysis of the type of veneering revealed significant differences between prostheses that had fully veneered and only vestibularly veneered double crowns. In correlation with the parameter “plaque index”, verstibularly veneered double crowns showed better results than fully veneered double crowns (*p* = 0.007). Probing depths ≥ 4 mm occurred significantly less frequently in partially veneered double crowns (13.6%) than in fully veneered double crowns (28.1%; *p* = 0.022). Likewise, caries and abutment defects, occurred approximately 50% less frequent at vestibularly veneered double crowns than in fully veneered double crowns (caries: 2.9% versus 5.5%, *p* = 0.366; biological complications: 6.4% versus 13.3%, *p* = 0.074).

### 3.8. Patient-Reported Outcome Measures

Sixty patients with 81 dentures answered the VAS questionnaires. Overall satisfaction with the NP-D-RPDs was 94.3 ± 9.4% (Figure 1). No statistically significant correlations could be found between the patient satisfaction with the NP-D-RPD and the following parameters: follow-up time-point, type of veneering, type of dental support (linear, triangular, and quadrangular), number of abutment teeth per NP-D-RPD, stability of the denture, and gender.

In detail, a one-way analysis of variance was performed to analyze whether overall patient satisfaction correlated with different parameters. No statistical difference was found for patients of the three recall groups (SF, MF, and LF) in their degree of overall satisfaction with the prostheses (SF: *n* = 20; mean: 94.8%; SD ± 8.3; MF: *n* = 46; mean: 94%; SD ± 9.2; LF: *n* = 14; mean: 96.2%; SD ± 5.9) (*p* = 0.7). The type of veneering had no influence on overall patient satisfaction: facial veneering (*n* = 40; mean: 93.4%; SD ± 8.7) versus full veneering (*n* = 40; mean: 95.7%; SD ± 8.1) (*p* = 0.220). The type of dental support did not affect patient satisfaction: linear support (*n* = 18; mean: 94.5%; SD ± 9.2) versus triangular support (*n* = 30; mean: 93.8%; SD ± 8.1) versus quadrangular support (*n* = 32; mean: 95.3%, SD ± 8.6) (*p* = 0.719). The number of abutment teeth did not influence overall patient satisfaction: dentures retained by two abutment teeth (*n* = 19; mean: 94.8%; SD ± 9.0), dentures retained by three abutment teeth (*n* = 29; mean: 93.6%, SD ± 8.1); dentures retained by four abutment teeth (*n* = 25; mean: 94.4%, SD ± 9.5), and dentures retained by five abutment teeth (*n* = 5; mean: 98.4; SD ± 2.1) showed no significant difference (*p* = 0.719). The stability of dentures did not affect overall patient satisfaction: sufficient stability (*n* = 70; mean: 94.6%; SD ± 8.4), moderate stability (*n* = 8; mean: 93.1%; SD ± 10.08; insufficient stability (*n* = 2; mean: 99%; SD ± 1.4) (*p* = 0.673). The overall satisfaction of men (*n* = 50; mean: 93.9%; SD ± 8.5) and women (*n* = 30; mean: 95.8%; SD ± 8.3) did not differ (*p* = 0.338).

## 4. Discussion

The aim of this cross-sectional in vivo investigation was to evaluate the survival and success rates of non-precious metal alloy NP-D-RPDs and the analysis of subjective patient satisfaction.

While the clinical performance of precious metal alloy double crown-retained removable partial dentures (P-D-RPDs) is documented for long-term observation periods of >5 years [10,11,28,29], only few data are available on non-precious metal alloy NP-D-RPDs. A systematic review on D-RPDs from 2011 included 11 studies that demonstrated survival rates between 60.6% and 95.3% after observation periods of 4–10 years [29]. In detail, the survival rates of tooth-supported D-RPDs were 90.0% and 95.1% after 4 and 5.3 years, respectively. Widbom et al. (2004) reported a survival rate of 96.7% with a mean observation period of 3.8 years [30]. Wöstmann et al. (2007) reported a survival rate of 95.1% for D-RDPs after a mean observation period of 5.3 years [10]. 

The survival rate of NP-D-RPDs in the present study was high, with 100% survival after the mean follow-up time of 2.1 ± 1.4 years. However, this observation time is short in comparison to existing studies. Nevertheless, to the best of the authors’ knowledge, only one other retrospective study exists on NP-D-RPDs. Zierden et al. (2018) calculated a 5-year survival rate of 95.9% for the NP group, with a mean observation time of 2.99 years [20]. In the mentioned study, 360 NP-D-RPDs with 1430 NP primary crowns were included, while in the present study only 82 NP-TRPDs with 268 NP primary crowns were evaluated. 

While the survival rate was high, the success rate of 68.3% was notably lower and decreased in the three follow-up groups (SF: 95.2%; MF: 63%; LF: 46.7%), demonstrating that NPA-D-RPDs came along with technical and biologic complications already shortly after the insertion of the prostheses and complications increased with the time after insertion. It is difficult to compare this success rate with the reported success rates in other studies, since success rates are rarely mentioned and, if they are, the definition of success differs substantially among publications. Comparable to the present study, Wagner and Kern counted minor necessary repairs as a complete success. The success rate was 41% after 10 years for conical crown-retained RPDs [11]. Wöstmann et al. reported on necessary treatments in 65% of all D-RPDs during the functional period, which can be interpreted as a success rate of 35% [10]. For overdentures supported by <4 telescopic abutment teeth, Rinke et al. found a success rate (event-free restorations) of 13% after 5 years [28], supporting the need for high maintenance and adaptation of (D-)RPDs shortly after insertion. In the present study, the chipping of the veneering of secondary crowns (9.8%) and the need for the renewal of prosthetic resin teeth (4.9%) were the main reasons for the rather low success rate at the prostheses level. This is in line with Zierden et al., showing a 10% need for the renewal of the veneering of secondary crowns, for NP- and P-D-RPDs. Others reported on the necessary re-veneering of secondary crowns in 9.3% to 26.9% of cases, one of the most frequent complications of D-RPDs [10,31,32]. Most likely, errors in the manufacturing process as well as stress occurring between the framework and the veneering material during insertion, removal and chewing cause the aforementioned fatigue of the veneering.

Further, the loss of denture retention over time seems to be a common issue. Zierden et al. described a necessary adjustment of the retention (increase) more often for precious metal alloy telescopic crowns than for non-precious metal alloy telescopic crowns (in total 4.2%), due to the higher material hardness of non-precious metal alloys. On the other hand, retention needs to be reduced for NPA-D-RPDs more often, which is in line with an in vitro study by Schimmel et al. [14]. They demonstrated stable retention forces between primary and secondary telescopic crowns made of NPA, but on an approximately 1.3-fold higher level compared to precious metal alloy telescopic crowns, which served as a control group in that study. Furthermore, the wear of the primary and secondary telescopic crowns was smaller with NPA than PA [14]. In the present study, an adjustment of retention was not documented. Nevertheless, denture stability was documented, and no insufficient stability was noticeable during the first year after insertion of the D-RPDs, whereas denture stability was found to be insufficient in one denture (2.2%) of the MF group and in one denture (6.7%) of the LF group. A loss of denture retention and denture stability over time seems to be a frequent complication [10,11] and is most likely induced by the wear of the contacting metal surfaces of the primary and secondary crowns during the insertion and removal of the D-RPDs. In the future, it will be of particular interest to assess how the retention behavior of NPA telescopic crowns will evolve over time, as the transition from traditional casting techniques towards CAD/CAM techniques continues. Existing in vitro studies have already shown that retention loss seems to be lower with milled crowns than with cast telescopic crowns [33]. Whether this can be transferred to in vivo studies has to be examined in the future.

In terms of the abutment tooth, a loss of sensitivity was the predominant complication. Of 268 abutment teeth, 66.8% were sensitive, 16.8% were endodontically treated, and 11.2% were non-sensitive. Zierden et al. (2018) reported a higher risk to be extracted for endodontically treated abutment teeth without post and core buildups than for vital abutment teeth [20]. Similarly, in the present study, D-RPDs with sensitive abutment teeth presented higher success rates (76.2%) than D-RPDs with at least one included non-sensitive abutment tooth that was not endodontically treated (43.8%). The loss of sensitivity of abutment teeth restored with telescopic crowns seems to be comparable to the loss of sensitivity of teeth restored with fixed partial dentures such as bridges or crowns. This is most likely the result of an irreversible trauma of the pulp due to chemical, mechanical and thermal irritations during the tooth preparation (grinding trauma) [34].

The prevalence of caries was rather low with 4.1% compared to other studies, reporting on 6% to 12.4% caries of abutment teeth after up to 10 years [11,30,35]. Therefore, it can be assumed that the susceptibility of abutment teeth of non-precious metal alloy telescopic prostheses to caries is comparable to or even lower than reported numbers of precious metal alloy telescopic prostheses. A possible bias of the present study may be that all patients were enrolled in a 6-month recall, and received a professional dental cleaning during every recall-visit, which may explain the low prevalence of caries.

In the present study, 2.2% of the 268 abutment teeth were lost after 2.1 ± 1.4 years. Zierden et al. reported 8.3% abutment tooth loss and, in general, abutment tooth loss ranged between 3% and 14.3%; however, this was after longer observation periods [20,30]. The presence of plaque seems to have an influence on abutment tooth loss, as patients with a high PI lost more abutment teeth (16.7%) compared to patients with low PI (0%) (*p* = 0.035). 

The prosthesis-related abutment tooth loss rate increased with the follow-up time, resulting in abutment tooth survival rates of 100%, 98% and 94.1%, for the SF, MF and LF groups, respectively. Similarly, there was an increase in the loss of sensitivity, the plaque index, the probing depth, Sulcus Bleeding Index, caries, and technical defects in all three groups of the present study. These results are in consistence with Makowski 2010 [34], who demonstrated similar results regarding bleeding on probing (53%). Widbom et al. (2004) reported on probing depths > 4 mm in 11.6% of the abutment teeth after a mean observation period of 3.8 years [30]. Coca et al. (2000) found an increase in probing depths in 54% of abutment teeth [36].

Regarding the distribution of abutment teeth, prostheses with a quadrangular support polygon had the highest success rate (69.7%); however, there was no statistically significant difference to linear (68.4%) or triangular (66.7%) supported prostheses. Nevertheless, quadrangular supported prostheses showed significantly more abutment teeth with a loss of sensitivity than triangular and linear supported prostheses. Hypothetically, an adaption of the abutment angulation during the preparation for common insertion path might have led to a pronounced loss of hard tissue and thus to an irritation of the pulp, resulting in a non-sensitive tooth over time. 

The highest success rate was found in Kennedy Class I patients, whereas Kennedy class II patients presented a significantly lower success rate of the D-RPDs (*p* = 0.003). Kennedy Class II patients presented significantly more dentures with abutment tooth loss (33.3%) compared to Kennedy Class I patients (*p* = 0.002). This can be explained by the possibly more favorable distribution of masticatory forces in Kennedy Class I cases compared to Kennedy Class II cases. Although, these results are in contrast to other studies, which could not find any significant influence of the Kennedy classes on the survival probability of the D-RPDs [10,20].

The number of abutment teeth per prosthesis had no statistically significant impact on the success of the prosthesis; however, the highest success rate was found in prostheses supported by five (80%), followed by four (73.1%), two (70%) and three (62.1%) abutment teeth. Likewise, Makowski reported no statistically significant correlation between success and the number of abutment teeth per prosthesis [34]. Zierden et al. reported no success rates; however, they showed lower survival rates for prostheses supported by one or two abutment teeth than prostheses supported by more than two abutment teeth. The 5-year survival rate for one abutment tooth prosthesis was 92.9%, for two 93.9%, for three 98.3% and for ≥four 95.2% [20]. Other studies reported 5-year survival rates of 79% and 93.6% for prostheses with two abutment teeth [37]. Weber (2005) documented higher survival rates with an increasing number of abutment teeth per prosthesis [38]. He explained his results by a better distribution of occlusal forces resulting in a lower load per abutment tooth and thus in a uniform redistribution of forces. Wenz et al. (2001) used two study groups. One group consisted of telescopic prosthesis with ≤3 abutment teeth; the second group included telescopic prosthesis with ≥4 abutment teeth. No statistically significant influence of the number of abutment teeth as a cofounding factor was reported [39].

Further, our results suggest a positive effect of partially veneered non-precious metal alloy telescopic crowns in comparison to fully veneered ones on the marginal gingiva. This is likely related to the smaller amount of veneering composite in intime contact to surrounding soft tissue, thus leading to less plaque accumulation and less gingival inflammation.

Regarding the use of precious versus non-precious metal alloys, Zierden et al. found no significant impact of the two types of alloys on T-RPD survival. Nevertheless, abutment tooth survival (10 years) was lower in non-precious alloy TRPDs compared to precious alloy TRPDs (67.7% versus 71.8%). On the other hand, more maintenance was necessary for precious alloy T-RPDs. Finally, costs can be reduced up to nearly 40% when using non-precious alloys. Therefore, the authors concluded that the use of non-precious alloy T-RPDs is a viable treatment option [20]. When it comes to the comparison of clasp-retained removable partial dentures (C-RPDs) and T-RPDs, Wagner and Kern compared the two designs and the combination of clasp-retention and conical crown retention 10 years after insertion [11]. In their study, they demonstrated higher failure rates in C-RPDs (66.7%) compared to T-RPDs (33.3%), or the combination of double-crowns and clasps (44.8%).

Besides the information on the long-term outcomes and complication rates, it seems important to know whether patients are satisfied with the results of a treatment. At the level of subjective patient satisfaction, there was very high acceptance of the incorporated prostheses for all investigated parameters. Ideally, a baseline evaluation of patient satisfaction before treatment would have been performed in addition to the follow-up questionnaire. Due to the retrospective design of the present study, this was not possible. Therefore, some negative outliers cannot be clearly explained. Interestingly, the overall satisfaction with the prostheses was high, with a mean value of 94.3%, even though other parameters such as pain, comfort, retention, and function showed negatively rated outliers. This implies that patients can be, overall, satisfied with their dentures, even though certain aspects of their dentures are not fully satisfactory to them. A mean value of 92.9% was given for satisfaction when speaking and 90.7% for the cleanability of the prostheses. Makowski et al. (2010) reported a satisfaction for the aesthetics of 96.9% and the cleanability of 100%. Comfort was rated at 87.5% (no feeling of tension) and 78.1% (good handling). The subjects’ satisfaction with the retention of the prostheses decreased over time. In the short-term group, satisfaction with retention was 94.6%, in the medium-term group 89.4% and in the long-term group 84.7%. This is in accordance with the objective findings. The chewing ability was rated at 84%. This is consistent with the reported satisfaction values of precious metal alloy telescopic crown prostheses [34]. Thus, when considering patient satisfaction with regard to the choice of metal alloy, there seems to be no noticeable difference between precious and non-precious metal alloys.

## 5. Conclusions

Within the limitations of the present study, it can be concluded that partially veneered non-precious metal alloy D-RPDs seem to be an efficient alternative to precious metal alloy D-RPDs. The subjective patient satisfaction exhibited excellent results. However, further studies should focus on long-term evaluations as well as cross-over designs with precious metal alloy D-RPDs and tooth-/implant-supported FPDs to obtain more reliable data on the efficiency of non-precious metal alloy D-RPDs.

## Figures and Tables

**Figure 1 materials-15-06137-f001:**
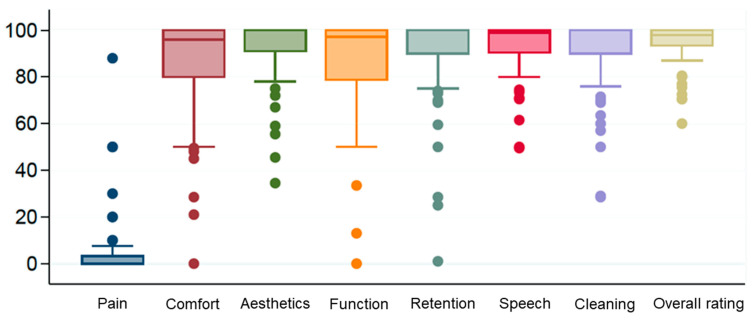
Patient-reported outcome measures of all recall groups regarding patients’ satisfaction for the different investigated parameters.

**Table 1 materials-15-06137-t001:** Summary of findings differentiated by the follow-up groups.

Group	Follow-Up Time, Months (Mean ± SD)	Abutment Teeth (*n*)	Dentures (*n*)	Abutment Survival	Prosthetic Survival	Prosthetic Succes
**Overall**	25.2 ± 16.5	268	82	96.4%	100%	68.3%
**Short** **Follow-up**	7.1 ± 3.4	65	21	100%	100%	95.2%
**Medium** **Follow-up**	24.9 ± 6.9	152	46	98%	100%	63%
**Long** **Follow-up**	51.7 ± 12.7	51	15	94.1%	100%	46.7%

**Table 2 materials-15-06137-t002:** Plaque index in relation to the presence of caries, Sulcus Bleeding Index (SBI) and probing pocket depth (PPD).

Plaque Index	0 (*n* = 87)	1 (*n* = 110)	2 (*n* = 54)	3 (*n* = 11)	*p*-Value
Prescence of caries	1.1%	2.7%	7.4%	27.3	0.631
SBI	34.5%	51.8%	77.8%	100%	<0.001
PPD ≥ 4 mm	5.7%	26.4%	24.1%	72.7%	<0.001

## Data Availability

Data supporting reported results can be retrieved from the Medical Center—University of Freiburg, Center for Dental Medicine, Department of Prosthetic Dentistry, Faculty of Medicine, University of Freiburg, Hugstetter Str. 55, 79106 Freiburg, Germany.

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
