# Peer review of "Non-Precious Metal Alloy Double Crown-Retained Removable Partial Dentures: A Cross-Sectional In Vivo Investigation"

_materials, 2022, doi:10.3390/ma15176137_

Round 1
Reviewer 1 Report
This cross-sectional, retrospective study is interesting, it has an appropriate and detailed design that allows the replication of the study and the results obtained reflect the applied methodology supported by the statistical analysis of easy interpretation.
I suggest continuing to investigate with randomized studies that demonstrate the efficiency and success rate between restored with precious and non-precious metal alloys double crown-retained removable partial dentures.
Author Response
Dear reviewer
Thank you for the evaluation of our manuscript.
We performed spell-checking, as suggested.
Reviewer 2 Report
Reviewed paper aims at analyzing clinical and patient-reported outcomes of using non-precious metal alloy double crown retained removable partial dentures. I find this work interesting and of high importance from the point of view of the question whether or not non-precious metal alloys might be effective substitute to precious metal alloy in making D-RPDs. However, I would like to rise several questions that should be answered by the Authors:
1. According to the Results (p. 5, from line 213) 61 patients were examined during the study, but adding the number of patients in the three subgroups (SF, MF and LF) it ends up with 65 patients. Is it possible that some patients wore dentures that allowed them to be counted in more than one group? Please, explain that.
2. All the numbers presented in the work should be given with the same precision in terms of significant digits. Now it looks like some results are given with 3 SDs (ca. 1 % precision), whereas some with only 2 SDs (ca. 10 % precision).
3. Patient outcomes recorded by the VAS questionnaire appear the most biased results presented in the paper. First of all, I do not feel too much difference between 'Comfort' and 'Overall satisfaction' measures, hence I think that all the terms should be explained deeper. Apart from that it turns out from Fig. 1 that there are patients who rate their sever pain, and low comfort, retention and function, but none of them rate the overall satisfaction below 60 %, which I find pretty much. I bet that these are not the same patients, so I suggest the Authors to present these plots in the form that will show connections between data from the same questionnaire.
Author Response
Thank you for evaluating our manuscript. Please see our replies to your comments. We applied all the changes indicated by the reviewer:
- According to the Results (p. 5, from line 213) 61 patients were examined during the study, but adding the number of patients in the three subgroups (SF, MF and LF) it ends up with 65 patients. Is it possible that some patients wore dentures that allowed them to be counted in more than one group? Please, explain that.
Answer: Thank you for the comment, yes, this is correct, 65 dentures were inserted into 61 patients. Four patients received two dentures at different timepoints because they were regular patients at our university clinics and throughout the time, they needed prostheses for upper and lower jaw, which was not an exclusion criteria of our study (see line page 2, line 97). We adapted the text to make this more clear to the reader ïƒ page 5, line 228-230
The text now reads: “In the present cross-sectional study, 61 patients (age: 65.6 ± 10.8 years, age range 39 y to 90 y) who received in total 65 dentures, were clinically and radiographically examined in 2016. Four patients received 2 dentures (upper and lower jaw) at different time points, whereas all other patients received only one denture.”
- All the numbers presented in the work should be given with the same precision in terms of significant digits. Now it looks like some results are given with 3 SDs (ca. 1 % precision), whereas some with only 2 SDs (ca. 10 % precision).
Answer: Thank you, we corrected this throughout the text
- Patient outcomes recorded by the VAS questionnaire appear the most biased results presented in the paper. First of all, I do not feel too much difference between 'Comfort' and 'Overall satisfaction' measures, hence I think that all the terms should be explained deeper. Apart from that it turns out from Fig. 1 that there are patients who rate their sever pain, and low comfort, retention and function, but none of them rate the overall satisfaction below 60 %, which I find pretty much. I bet that these are not the same patients, so I suggest the Authors to present these plots in the form that will show connections between data from the same questionnaire.
Answer: Thank you for this important point.
To make the difference more clear between comfort and overall satisfaction, we added to the text (page 4, lines 175-181) : “It was further explained to the patients, that “comfort” referred to wearing comfort, asking the patients to evaluate how comfortable the denture felt during function and in resting position (e.g. whether the patients felt any pressure spots that were not painful, yet uncomfortable). Whereas “overall satisfaction” referred to the personal satisfaction of the patients, when considering factors, such as comfort, aesthetics, function, cleanability, and other aspects”
In view of the reviewer's comment, the outliers in the questionnaires were once again considered specifically. Since the negative outliers were found, in both, questionnaires of the same patients and in individual questions in patients who otherwise had only very high ratings, a graphical presentation seems difficult. Ideally, a baseline evaluation before treatment and an evaluation after treatment would have been combined. However, this is not possible due to the retrospective study design. For these two reasons, even after consultation with the statistician, we could not find a solution for a better graphical representation of the results of the individual questionnaires. We have included this problem in the discussion as a limitation in the interpretation of the results (ïƒ page 11, lines 618-626).
The text now reads:
«Ideally, a baseline evaluation of patient satisfaction before treatment would have been performed in addition to the follow-up questionnaire. Due to the retrospective design of the present study, this was not possible. Therefore, some negative outliers cannot be clearly explained. Interestingly, the overall satisfaction with the prostheses was high, with a mean value of 94.3%, even though certain other parameters such as pain, comfort, retention, and function showed negatively rated outliers. This implies, that patients can be overall satisfied with their dentures, even though certain aspects of their dentures are not fully satisfactory to them.»
Reviewer 3 Report
The paper is well written.
It might be of interest to describe the distribution of abutment teeth and did it effected the success rate. ?
Author Response
Thank you for evaluating our study. We addressed all your comments:
It might be of interest to describe the distribution of abutment teeth and did it effected the success rate. ?
Answer: Thank you for this comment, we ran statistics on the distribution of abutment teeth and could not find any correlation between abutment tooth distribution and success rate. Please find the respective discussion on these results on page 10, lines 568-575.
Reviewer 4 Report
Comments to the Author
The paper entitled “Non-precious metal alloy double crown-retained removable partial dentures: a cross-sectional in vivo investigation” has the aim to analyze the clinical and patient-reported outcomes of patients restored with non-precious metal alloy double crown retained removable partial dentures (NP-D-RPDs). The manuscript contains the new results. I have the following comments and hope these comments could help authors further refine their research:
My minor comments are as follows.
1) Page 2, Line 83-84
These patients had been treated by undergraduate students who were constantly supervised by specialized prosthodontists.
> The accuracy of abutment tooth formation and NP-D-RPD fabrication has a significant impact on prognosis. The text states that the NP-D-RPDs were created by graduate students during a specialized prosthetic course, but has there been any Inter-operator calibration or post-training checks by the instructor?
If calibration has been performed, it should be included in the text.
2) Page 5, Line 228
Technical complications occurred in 22% (n=18) of the dentures.
> Is the occurrence of technical complications in 22% of dentures more common than in previous studies and clinical statistics?
Are the results similar for clasp and precious metal dentures?
Page 5, Line 239
Out of a total of 268 abutment teeth, secondary caries was found in 4.1% (n=11).
Page 10, Line 432-433
The prevalence of caries was rather low with 4.1% compared to other studies, report-432 ing on 6% to 12.4% caries of abutment teeth after up to 10 years [11,30,35].
>The extremely low prevalence of dental caries compared to previous studies may indicate that the patients in this study have high dental IQ and interest in dentistry.
Could this be a bias factor?
Author Response
Thank you for evaluating our study. We addressed all your comments:
1) Page 2, Line 83-84
These patients had been treated by undergraduate students who were constantly supervised by specialized prosthodontists.
> The accuracy of abutment tooth formation and NP-D-RPD fabrication has a significant impact on prognosis. The text states that the NP-D-RPDs were created by graduate students during a specialized prosthetic course, but has there been any Inter-operator calibration or post-training checks by the instructor?
If calibration has been performed, it should be included in the text.
Answer: We agree, that the accuracy of abutment tooth formation and NP-D-RPD fabrication has a significant impact on prognosis. As students were constantly supervised by specialized prosthodontists, every key intermediate stage had to be validated by the supervisors. Those supervisors were doing the practical student training since several years in the same constellation, and had been calibrated throughout the years by a very clearly defined protocol, established by the head of the department.
We added some information to the text: page 2, lines 84-87.
The text now reads: “These patients had been treated by undergraduate students who were constantly supervised by the same four specialized prosthodontists.The supervisors had undergone calibration prior to the student education course, and adhered to a clearly defined protocol established by the head of the department. All key intermediate steps during the treatment had to be validated by the supervisors.”
2) Page 5, Line 228
Technical complications occurred in 22% (n=18) of the dentures.
> Is the occurrence of technical complications in 22% of dentures more common than in previous studies and clinical statistics?
Answer: Unfortunately, this question is rather difficult to answer, due to heterogeneous way of reporting on survival/success/complications amongst the publications. We do not find the complication rates clearly stated in literature, but instead, success and survival rates are given. For example, in Schwindling et al.’s study, success is defined as “survival without severe complications”. In this case, success rates of T-RPDs (telescopic crown retained RPDs), C-RDPs(conical crown retained RPDs) and R-RDPs (resilient telescopic crown retained overdentures) ranged between 90-78.5% after 7 years – which means, that 10-21.5% of the prostheses were not successful, meaning an occurrence of severe complications. It is not described in detail though, which kind of complications occurred in which amount, which makes it difficult to compare the data to our outcomes. In the discussion, we nevertheless compared outcomes of different studies the best way we could – pointing out certain complications and their occurrence rates, as for example chipping, which is mentioned in Zierden et al. and Wöstmann et al. (page 9, line 487-507) Our results on chipping rates (9.8%) are very similar to the outcomes of Zierden et al. with 10% need for renewal of the veneering (page 9, line 499-507)
Are the results similar for clasp and precious metal dentures?
Answer: We added two new paragraphs, addressing the comparison of precious versus non-precious alloys, and clasp- versus telescopic crown retained partial dentures, even though we have no data on this topic ourselves. (page 11, lines 598-613).
The text now reads: «Regarding the use of precious versus non-precious alloys, Zierden et al. found no significant impact of the two types of alloys on T-RPD survival. Nevertheless, abutment tooth survival (10 years) was lower in non-precious alloy TRPDs compared to precious alloy TRPDs (67.7% vs 71.8%). On the other hand, more maintenance was necessary for precious alloy T-RPDs. Finally, costs can be reduced up to nearly 40% when using non-precious alloys. Therefore, the authors concluded, that the use of non-precious alloy T-RPDs is a viable treatment option.» (page 11, lines 569-575).
And (page 11, lines 575-579): “When it comes to the comparison of clasp-retained removable partial dentures (C-RPDs) and T-RPDs, Wagner and Kern compared the two designs and the combination of clasp-retention and conical crown retention 10 years after insertion [11]. In their study, they demonstrated higher failure rates in C-RPDs (66.7%) compared to T-RPDs (33.3%), or the combination of double-crowns and clasps (44.8%).”
Page 5, Line 239
Out of a total of 268 abutment teeth, secondary caries was found in 4.1% (n=11).
Page 10, Line 432-433
The prevalence of caries was rather low with 4.1% compared to other studies, report-432 ing on 6% to 12.4% caries of abutment teeth after up to 10 years [11,30,35].
>The extremely low prevalence of dental caries compared to previous studies may indicate that the patients in this study have high dental IQ and interest in dentistry.
Could this be a bias factor?
Answer: As all patients were enrolled in a 6-month recall, and received a professional dental cleaning during every recall-visit, this may explain the very low prevalence of caries. We added this information into the text. Page 10, lines 541-543.
It now reads: “A possible bias of the present study may be, that all patients were enrolled in a 6-month recall, and received a professional dental cleaning during every recall-visit, which may explain the low prevalence of caries.”